# Optimizing Genetic Workup in Pheochromocytoma and Paraganglioma by Integrating Diagnostic and Research Approaches

**DOI:** 10.3390/cancers11060809

**Published:** 2019-06-11

**Authors:** Laura Gieldon, Doreen William, Karl Hackmann, Winnie Jahn, Arne Jahn, Johannes Wagner, Andreas Rump, Nicole Bechmann, Svenja Nölting, Thomas Knösel, Volker Gudziol, Georgiana Constantinescu, Jimmy Masjkur, Felix Beuschlein, Henri JLM Timmers, Letizia Canu, Karel Pacak, Mercedes Robledo, Daniela Aust, Evelin Schröck, Graeme Eisenhofer, Susan Richter, Barbara Klink

**Affiliations:** 1Institute for Clinical Genetics, Medical Faculty Carl Gustav Carus, Technische Universität Dresden, 01307 Dresden, Germany; Laura.Gieldon@tu-dresden.de (L.G.); Karl.Hackmann@uniklinikum-dresden.de (K.H.); Arne.Jahn@uniklinikum-dresden.de (A.J.); Johannes.Wagner@uniklinikum-dresden.de (J.W.); Andreas.Rump@uniklinikum-dresden.de (A.R.); Evelin.Schroeck@uniklinikum-dresden.de (E.S.); 2Core Unit for Molecular Tumor Diagnostics (CMTD), National Center for Tumor Diseases (NCT), 01307 Dresden, Germany; doreen.william@nct-dresden.de (D.W.); w.jahn@dkfz-heidelberg.de (W.J.); 3German Cancer Consortium (DKTK), 01307 Dresden, Germany; 4German Cancer Research Center (DKFZ), 69120 Heidelberg, Germany; 5Institute of Clinical Chemistry and Laboratory Medicine, University Hospital Carl Gustav Carus, Medical Faculty Carl Gustav Carus, Technische Universität Dresden, 01307 Dresden, Germany; Nicole.Bechmann@uniklinikum-dresden.de (N.B.); Graeme.Eisenhofer@uniklinikum-dresden.de (G.E.); Susan.Richter@uniklinikum-dresden.de (S.R.); 6Medizinische Klinik und Poliklinik IV, Klinikum der Universität, LMU München, 80336 Munich, Germany; Svenja.Noelting@med.uni-muenchen.de; 7Institute of Pathology, Ludwig-Maximilians-University, 80337 Munich, Germany; Thomas.Knoesel@med.uni-muenchen.de; 8Department of Otorhinolaryngology Head and Neck Surgery, Municipal Hospital Dresden, 01067 Dresden, Germany; Volker.Gudziol@uniklinikum-dresden.de (V.G.); 9Department of Internal Medicine III, University Hospital Carl Gustav Carus at Technische Universität Dresden, 01307 Dresden, Germany; Georgiana.Constantinescu@uniklinikum-dresden.de (G.C.), Jimmy.Masjkur@med.uni-heidelberg.de (J.M.); 10Klinik für Endokrinologie, Diabetologie und Klinische Ernährung, Universitätsspital Zürich, 8091 Zürich, Switzerland; felix.beuschlein@usz.ch; 11Department of Internal Medicine, Radboud University Medical Centre, 6525 Nijmegen, The Netherlands; henri.timmers@radboudumc.nl; 12Department of Experimental and Clinical Biomedical Sciences, University of Florence, 50149 Florence, Italy; letizia.canu@unifi.it; 13Eunice Kennedy Shriver National Institute of Child Health and Human Development, National Institutes of Health, Bethesda, MD 20892, USA; karel@mail.nih.gov; 14Hereditary Endocrine Cancer Group, CNIO, Madrid, Spain and Centro de Investigación Biomédica en Red de Enfermedades Raras (CIBERER), 28029 Madrid, Spain; mrobledo@cnio.es; 15Institute of Pathology, Tumor and Normal Tissue Bank of the UCC/NCT Dresden, University Hospital Carl Gustav Carus, Technische Universität Dresden, 01307 Dresden, Germany; Daniela.Aust@uniklinikum-dresden.de; 16National Center for Genetics (NCG), Laboratoire national de santé (LNS), 1, rue Louis Rech, 3555 Dudelange, Luxembourg

**Keywords:** pheochromocytoma, paraganglioma, next-generation sequencing, sporadic, hereditary, CNV detection

## Abstract

Pheochromocytomas and paragangliomas (PPGL) are rare neuroendocrine tumors with a strong hereditary background and a large genetic heterogeneity. Identification of the underlying genetic cause is crucial for the management of patients and their families as it aids differentiation between hereditary and sporadic cases. To improve diagnostics and clinical management we tailored an enrichment based comprehensive multi-gene next generation sequencing panel applicable to both analyses of tumor tissue and blood samples. We applied this panel to tumor samples and compared its performance to our current routine diagnostic approach. Routine diagnostic sequencing of 11 PPGL susceptibility genes was applied to blood samples of 65 unselected PPGL patients at a single center in Dresden, Germany. Predisposing germline mutations were identified in 19 (29.2%) patients. Analyses of 28 PPGL tumor tissues using the dedicated PPGL panel revealed pathogenic or likely pathogenic variants in known PPGL susceptibility genes in 21 (75%) cases, including mutations in *IDH2, ATRX* and *HRAS*. These mutations suggest sporadic tumor development. Our results imply a diagnostic benefit from extended molecular tumor testing of PPGLs and consequent improvement of patient management. The approach is promising for determination of prognostic biomarkers that support therapeutic decision-making.

## 1. Introduction

Pheochromocytomas and paragangliomas (PPGL) are rare neuroendocrine tumors that originate from neural crest-derived chromaffin cells and develop either in the adrenal medulla or in extra-adrenal sympathetic and parasympathetic ganglia. PPGLs show the highest heritability of all cancers with almost 40% of patients carrying pathogenic germline mutations in one of the known PPGL susceptibility genes. [1,2,3,4,5] For this reason, current guidelines recommend that germline genetic testing should be considered in all patients with PPGLs, regardless of family history and age at diagnosis [6]. Identification of a predisposing germline variant enables predictive testing in PPGL families, clinical surveillance of healthy mutation carriers and risk stratification for malignant disease and for development of synchronous and metachronous tumors. 

The majority of hereditary PPGLs are caused by mutations affecting *NF1*, *RET, VHL, SDHA, SDHB, SDHC* and *SDHD*. Rare underlying germline mutations have been described in *FH, MAX, SDHAF2* and *TMEM127.* [2,7,8,9,10,11]. Genetic testing of these genes in PPGL patients in routine diagnostics has previously been conducted according to a step-wise diagnostic algorithm [2]. With next generation sequencing techniques becoming cheaper and more commonly available, multi-gene panel sequencing is increasingly replacing targeted approaches. Recently, several additional susceptibility and candidate genes accounting for a small proportion of cases have been identified. These are, however, not yet commonly included in routine diagnostic analyses of affected patients. These include germline mutations in genes encoding components of metabolic pathways, e.g. *MDH2*, *GOT2* and *DLST* [12,13,14] and in hypoxia pathway related genes *EGLN2* (*PHD1*), *EGLN1* (*PHD2*), and *EPAS1 (HIF2α)* [15,16]. Mutations in the latter usually occur somatically but can be associated with a syndromic presentation when they occur in mosaic forms [17]. Similarly, a case with a postzygotic mosaic mutation in *H3F3A* presenting with PPGLs and a giant cell tumor of the bone has recently been described [18].

Recent efforts in genomic analyses of PPGLs revealed that about 30% of tumors carry somatic mutations in known susceptibility and driver genes [19]. Furthermore, somatic mutations in known cancer associated genes have been identified as driver-alterations in a subset of PPGLs due to recent efforts in genome sequencing projects, such as somatic activating hot-spot cases, and somatic hot-spot mutations in *IDH1* or *IDH2,* recognized as drivers of tumorigenesis in PPGL, were identified [20,21]. In addition, somatic alterations in other cancer-associated genes, such as *SETD2, EZH2, FGFR1, BRAF, MET,* and *TP53,* have been reported [18,19,22]. 

Supporting analyses of tumor tissue by immunohistochemistry and metabolite measurements provide further information, aiding in identification and interpretation of genetic variants [21]. We previously demonstrated that quantification of Krebs cycle metabolites in tumor tissue can classify PPGLs and identify tumors with underlying alteration in *SDH-*genes (*SDHx*), *FH*, and *IDH-*genes *(IDHx)* and can furthermore aid in classification of variants of unknown significance in *SDHx* [21,23].

To improve PPGL diagnostics and patient management we developed a comprehensive customized PPGL multi-gene panel for the analysis of PPGL tumor tissue, comprising 84 PPGL associated and candidate genes. We applied this to a cohort of 28 PPGL tumor samples and compared the results to our in-house PPGL cohort of patients who, over the course of several years, have undergone germline sequencing within a routine diagnostic setting.

## 2. Results

### 2.1. Routine Germline Testing in PPGL Patients Solves 30% of Cases

Between 2008 and 2017, 65 patients with PPGLs were referred to the Institute of Clinical Genetics in Dresden for genetic counselling and/or genetic testing. Thirty-one patients primarily presented with pheochromocytoma (PHEO) and 34 patients with paraganglioma (PGL), of which 25 were head and neck paragangliomas (HNP). The median age at diagnosis was 46.5 years (ranging from 13 to 77 years) and did not differ between those patients diagnosed with pheochromocytoma and those with paraganglioma. Eleven patients (17.2%) had developed PPGL until 30 years of age (y), while 27 patients had an age of onset over 50y (41.5%). Forty-two patients (64.6%) were female (18 PHEOs, 24 PGLs) and 23 patients (35.4%) were male (13 PHEOs, 10 PGLs). 

Five of these patients (7.7%) had multiple PPGL tumors and 8 patients (12.3%) had additional tumors including adrenocortical adenoma/carcinoma, thyroid carcinoma, renal cell carcinoma, a testicular tumor, mamma carcinoma, Hodgkin’s lymphoma and cervical carcinoma. Pedigree information was available in 53 of the patients and inconspicuous regarding relatives with tumors in 34 of them (64.2%). Three patients had relatives affected by paraganglioma, pheochromocytoma or gastrointestinal stromal tumor (GIST), indicating hereditary disease in these families, which was molecularly confirmed in all three cases. One family was known to carry a pathogenic *BRCA2* germline mutation causing hereditary breast and ovarian cancer. In addition, pedigree analysis of 3 further families was suspicious of hereditary breast and ovarian cancer, hereditary gastric cancer and hereditary colon cancer, respectively. Another 15 families were affected by additional cancers unrelated to PPGL and without fulfillment of criteria for a specific hereditary tumor predisposing syndrome. Clinical data are summarized in Appendix A.

Germline testing of 11 PPGL susceptibility genes associated with known hereditary tumor predisposition syndromes associated with PPGLs (*SDHA, SDHB, SDHC, SDHD, SDHAF2, MAX, FH, NF1, RET, TMEM127* and *VHL*) was done in all of these patients, in the majority of cases using a combination of targeted multi-gene sequencing with the TruSight Cancer panel (Illumina, Inc., San Diego, CA, USA) in combination with customized array-based Comparative Genomic Hybridization (array-CGH) for copy number variation (CNV) calling [24] and Sanger Sequencing of *SDHA* (for details see Appendix A) 

In 19 of 65 cases (29.2%) we identified pathogenic or likely pathogenic germline mutations in a PPGL susceptibility gene, confirming a PPGL related hereditary tumor predisposition syndrome in these cases (Table 1). The majority (14) of these patients had germline mutations in an *SDHx* gene, which is associated with an aberrant succinate to fumarate ratio (S:F ratio) in the tumor tissue. In eight cases with pathogenic germline *SDHx* variants, succinate and fumarate concentrations in the tumors were analyzed and aberrant S:F ratios were evident in six of these cases (Table 1).

In one patient (ID3) with a pathogenic *SDHB* variant, we found an additional pathogenic germline variant in *CHEK2* (c.1100delC, p.(Thr367fs)) that is known to be associated with hereditary breast and ovarian cancer (Table 1). In this case, an aberrant S:F ratio was not observed in the tumor although a pathogenic *SDHB* variant was identified, which might be explained by reported inconsistencies between *SDHx* mutations and aberrant S:F ratios in head and neck paragangliomas (HNPs) [21].

### 2.2. Development of a Dedicated PPGL Custom Panel for Germline and Tumor Testing

We designed a custom PPGL panel comprising a total of 84 genes, including 20 well-defined PPGL susceptibility genes (*EGLN1, EGLN2, MDH2, FH, SDHA, SDHB, SDHC, SDHD, SDHAF2, MAX, RET, TMEM127, VHL, EPAS1, NF1, H3F3A, IDH1, IDH2, ATRX,* and *HRAS*) that have been described to occur as germline, somatic or mosaic mutations in PPGL [2,4]. Additionally, we included further common known tumor genes such as *TP53, PTEN, FGFR1,* and *BRAF* that have been described to be mutated in PPGL tumors and might be secondary mutations [18,19,22]. Furthermore, we included candidate genes for PPGLs based on their gene function or gene family, e.g., genes encoding for metabolic enzymes or genes involved in epigenetic regulation (such as *TET1* and *TET2*) (Figure 1, Appendix A). Some of these candidate genes have already been implicated to play a role in PPGL development, such as *KIF1B*, *GOT2,* and *IDH3B* [13,25].

Importantly, we chose an enrichment-based protocol to enable even nucleotide coverages across target regions and to limit coverage fluctuations as best as possible. Even overall coverage enables robust CNV detection using next generation sequencing (NGS) data. Furthermore, the enrichment-based design can be adapted to be used, not only for high-quality DNA, but also for low-quality and fragmented DNA from paraffin-embedded (FFPE) tumor tissues. [26]

### 2.3. Comprehensive Tumor Testing Improves Detection Rate of Underlying Mutations in PPGL-Patients

We first applied our novel customized PPGL panel to available tumor tissues of ten cases from our Dresden cohort that had undergone routine diagnostic germline testing. Four of the analyzed tumors were FFPE samples and six were freshly cryo-conserved tissue samples. Two of the cases (ID42 and ID43) had pathogenic germline variants in known PPGL susceptibility genes (*SDHB, SDHC*), whereas in eight cases, no pathogenic variants had been identified during routine germline testing (Table 2). Sequencing of tumor tissue from the aforementioned ten patients led to the identification of pathogenic variants in known PPGL susceptibility genes in nine cases (Table 2). In the two cases with pathogenic germline mutations, we could confirm both mutations in tumor tissue. In one of these cases (ID42), we observed increased allele frequency (85.4%) of the pathogenic *SDHB* variant, suggesting a loss of heterozygosity in this tumor (Table 2). In the other case (ID43) with a pathogenic germline variant in *SDHC*, we found an additional pathogenic somatic variant in *ATRX* (Table 2). In seven out of eight cases with inconspicuous germline analysis results, we identified pathogenic somatic variants in tumor tissue of the patients (Table 2). Four of these cases (ID69, ID66, ID67 and ID24) had a somatic variant in a known PPGL susceptibility gene (*SDHB*, *SDHD*, and 2x *VHL*) that was not present in the matched blood sample, indicating a sporadic tumor development in these cases. The tumor ID69 with a pathogenic *SDHD* variant additionally harbored a somatic missense variant of unknown significance in *FH* (p.(Ala198Val)) with a low allele frequency (7.3%), indicating a potential secondary event. Pathogenic hot-spot variants in *HRAS* were identified in two cases (ID1 and ID68). In the seventh tumor (ID72), we found pathogenic variants in *TP53* and in *ATRX*. In all of these cases, the family history was inconspicuous and, to our knowledge, no other tumors were diagnosed in those patients. All cases with pathogenic variants in *SDHx* also had aberrant S:F ratios in tumor tissue, whereas cases with pathogenic variants in non-*SDHx* genes did not show elevated S:F ratios, demonstrating correlation between genomic and metabolomic analyses (Table 2). Altogether, combined tumor and germline testing revealed causative mutations in 90% of the cases (9/10), confirming a sporadic tumor in seven cases. 

Next, we applied our PPGL custom panel to 18 PPGL tumor samples (twelve freshly cryo-conserved and six FFPE samples) within the scope of a multi-center research project. This cohort consisted of twelve pheochromocytomas and six paragangliomas, two of which were head and neck paragangliomas. 

In twelve of these 18 tumors, pathogenic variants in a known PPGL susceptibility gene were identified by PPGL custom panel sequencing (Table 2). In four cases, germline status of pathogenic variants could be evaluated by targeted Sanger sequencing. In two cases (ID51, ID71), we found pathogenic variants in *SDHB*. The pathogenic nonsense mutation p.(Tyr61*) in *SDHB* in case ID51 was confirmed to be a somatic variant [21]. Furthermore, metabolome analysis of tumor tissue of case ID51 showed an elevated S:F ratio of 5178.2, which is concurrent with the pathogenic *SDHB* variant [21]. The *SDHB* missense variant p.(Arg242Cys) in case ID71 showed a low allele frequency of 16.4%, indicating that this variant is likely somatic as well. However, in this case no elevated S:F ratio was detected in tumor tissue, which is likely attributable to known deviations between pathogenic *SDHx* variants and S:F ratio in head and neck paragangliomas [21]. Pathogenic variants in *FH* were identified in tumor tissue of two cases (ID41 and ID82) and showed elevated fumarate to malate ratios, demonstrating correlation between genomic and metabolomic analyses [21]. Both pathogenic *FH* variants were also identified in the germline of the patients by targeted analysis (Table 2) [21].

In four cases, we identified pathogenic variants in *NF1* (ID73, ID79, ID91 and ID92). One of these cases (ID92) had two different nonsense mutations in *NF1* (p.(Arg440*) and p.(Arg2517*)) at allele frequencies of 15.9% and 33.2%, respectively, which could also indicate somatic origin. Case ID73 had a *NF1* nonsense variant (p.(Gln514*) at an allele frequency of 62.1%, case ID79 showed a pathogenic frameshift variant (p.(Ser2601fs)) in *NF1* with an allele frequency of 83.2% and ID91 had a pathogenic splice variant in *NF1* with an allele frequency of 39.9%. A pathogenic missense variant in *VHL* (p.(Arg167Gln) was identified in ID78 at an allele frequency of 49.6%. No blood samples were available for analysis in these five cases and thus, presence of the respective *NF1* or *VHL* variants in the germline of these patients could not be determined.

One tumor (ID75) harbored a somatic *IDH2* hot-spot mutation, in line with elevated D-2-hydroxyglutarate [21]. Tumor ID60 presented with a known hot-spot mutation in *HRAS*. Although no germline sequencing data were available, it can be assumed that this hot-spot mutation was somatic based on the gene involved and the low allele frequency of the variant. In one tumor (ID80) we identified a nonsense mutation in ATRX (p.Glu481*), however, only at a frequency of 5.4%. This might imply the alteration was a secondary event contributing to tumor maintenance, although no other pathogenic variants were found in this tumor using the PPGL custom panel. 

Taken together, by sequencing analyses of altogether 28 tumor tissues using our PPGL panel, we identified 24 (likely) pathogenic variants in 21 tumor samples (Table 2, Figure 2). The spectrum of genes involved in these 21 samples differs from the spectrum of mutated genes identified by routine germline testing in our clinical cohort of 65 patients (Figure 2b). For example, the majority of variants identified by germline testing involved *SDHx* genes, while *SDHx* mutations only account for about 30% of mutations found by tumor testing (14/19 germline variants in the clinical cohort compared to 7/24 variants in tumor tissues). In contrast, variants in *IDH2*, *TP53*, *ATRX*, *HRAS*, and *VHL* were exclusively found by tumor sequencing, and variants in *NF1* were more frequently found in tumor tissues (5/24 variants in 21 tumor cases) compared to germline testing (1/19 variants identified in our clinical cohort of 65 patients (Figure 2b). 

In seven of 28 analyzed PPGL tumor samples, we found no underlying pathogenic variants in any of the 20 known PPGL susceptibility genes. In two of these seven cases, we identified potentially disease relevant variants. One tumor (ID61) had a missense variant in *ATRX* (p.(Asn53His)) at an allele frequency of 25%. This variant was not listed in databases such as gnomAD, ClinVar or dbSNP and has, to the best of our knowledge, not described in the literature [27,28,29]. In the tumor of patient ID90 we identified a missense variant in the candidate gene *TET1* (p.(Val128Leu)) with an allele frequency of 26.1%. The variant was found two times in heterozygous state in the general population (gnomAD database) and is listed in dbSNP (rs142008363), but not in ClinVar. Based on current knowledge, we classified both variants as variants of unknown significance. Unfortunately, blood samples to confirm the somatic status were unavailable in both cases, although the allele frequencies indicate that these variants might be somatic. 

### 2.4. Tumor Testing with Our PPGL Custom Panel Can Provide Additional Information about Secondary Somatic Changes

To gain additional information about further potentially pathogenic alterations, we performed NGS-based copy number variant (CNV) detection in all samples that were sequenced with the PPGL custom panel. A blood sample with a confirmed deletion of one *NF1* allele served as positive control for CNV detection and the *NF1* loss was reliably detected by the applied algorithms (Figure 3F).

Within our cohort of 28 PPGL tumors that were analyzed by custom panel sequencing, several pathogenic variants were found at high allele frequencies indicating a loss of heterozygosity (Table 2). In two cases with pathogenic *SDHB* variants (ID42: frequency of 85.4% and ID51: frequency of 80%) loss of one *SDHB* allele was identified by CNV detection analysis (Figure 3a,b). Similarly, pathogenic variants at high allele frequencies in *FH* were found in ID41 (92.3%) and ID82 (82%). Neither of these cases showed an *FH* deletion, supporting a copy number neutral mechanism of loss of heterozygosity (LOH) (Figure 3C, Appendix A).

Two tumors with inconspicuous sequencing results (ID76 and ID88) had high frequencies of CNVs compared to the other samples (Appendix A), including heterozygous loss of *TP53* in one tumor (ID76) and of *NF1* in the other tumor (ID88) (Figure 3D,E).

### 2.5. Identification of Variants in Candidate Genes 

Sequencing of PPGL tumor tissue with the custom panel revealed rare variants in candidate genes in nine cases (Table 3). All of these variants were missense variants and the significance of these variants is currently unknown. Two of these variants were found at low allele frequencies (ID61: *ATRX* p.(Asn53His) 25.2% and ID88: *TET1* p.(Val128Leu) 26.1%) in tumors that otherwise did not harbor a clear pathogenic or likely pathogenic mutation in a known PPGL susceptibility gene (see Section Section 2.3).

The other seven variants were identified in cases that also carried a pathogenic variant in a PPGL susceptibility gene and occurred at allele frequencies between 46% and 52.3% with the exception of *FH* p.(Ala198Val) with a frequency of 7.3% (Table 3). 

All seven variants were predicted to be pathogenic by several *in silico* prediction programs (ID41: *OGDHL* p.(Tyr447Cys), ID68: *GPT* p.(Glu210Cys), ID69: *FH* p.(Ala198Val), ID71: *PDHB* p.(Val174Met), ID78: *FGFR1* p.(Pro702Tyr), ID79: *HIST1H3B* p.(Pro44Gln) and ID82: *PCK2* p.(Arg155Cys), Table 3). None of the selected variants was listed in ClinVar or occurred at a significant frequency in the general population (gnomAD) [27,28]. Five of these variants were listed in dbSNP and two variants were identified as somatic variants in lung cancer (*FGFR1* p.(Pro702Tyr)) or in a colon carcinoma ((*GPT* p.(Glu210Cys), COSMIC database; (Table 3)) [29,30]. 

Since these variants occurred together with clearly pathogenic variants and (with the exception of the *FH* variant) with an allele frequency of about 50%, they might be rare/private germline variants that are coincidental findings. Unfortunately, blood samples of these nine cases were not available to confirm or exclude the somatic status of these variants.

## 3. Discussion

PPGLs are rare but known to be the tumor type with the highest heritability, with 30–40% of patients carrying a predisposing germline mutation. A steadily growing number of 20 PPGL susceptibility genes has been identified to date, and several candidate genes are being investigated [2,5]. Some of these genes have been found to be mutated both in the germline and somatically, while others only occur as somatic mutations [7,8,13,16,17,18,19,22,31,32].

Analyses of both tumor and blood-derived DNA therefore aid in discrimination of sporadic from hereditary tumor forms [31]. This information is crucial for stratifying the risk of synchronous or metachronous tumor development in PPGL patients both regarding additional PPGLs and regarding other tumor types that, respectively, are associated with some of the PPGL susceptibility genes such as gastrointestinal stromal tumors (GIST ) with *SDH*x mutations [33].

In some patients, occurrence of PPGL is the first manifestation of a heritable syndrome such as Neurofibromatosis type 1 (NF1) that besides a tumor predisposition is usually accompanied by additional symptoms [34,35]. Patients with hereditary conditions need to undergo genetic counselling and predictive testing can be offered to healthy family members. Healthy mutation carriers should then be included in tailored clinical surveillance programs for early detection of tumor development or other manifestations of disease. Depending on the gene affected, predictive testing can even be indicated in children and prenatal testing needs to be considered for severe conditions with highly variable phenotypic expression.

On the other hand, knowledge of somatic mutations can also be crucial for patient treatment and follow-up. Patients with pathogenic mutations in *SDHB* and *FH*, for example, have an elevated risk for malignant tumor development, regardless of whether the mutation was inherited or whether it occurred somatically [10,36]. Diagnosis of malignancy of PPGLs can, with certainty, only be made based on detection of metastases and not histology [37]. Therefore, patients with mutations associated with malignancy of tumors need to be followed up closely for metastatic disease. Moreover, knowledge of mutations, both somatic and germline, increasingly opens up the possibility for targeted therapeutic approaches [38].

Taking these implications on clinical management into account, it is evident that patients can substantially benefit from combined genetic analysis of both blood and tumor-derived DNA. We found that in our cohort germline testing identified pathogenic mutations in PPGL core genes in 19 out of 65 patients (29.2%). This number complies with reported germline mutational rates of 30–40% in PPGL patients in the literature [1,2,3,4,31]. Tumor analysis of 28 samples using the custom tailored PPGL panel revealed pathogenic mutations in 21 tumors (75%) of which 13 were available for germline testing (9 were from our routine clinical cohort who had received routine diagnostic testing, and 4 cases from our research cohort received targeted blood sequencing of variants after tumor testing, see Table 2). About one third of these 13 mutations found in tumor tissue were also present in blood and therefore presumed to be germline, while two third of these pathogenic mutations had developed somatically. Aim et al. recently published similar results with a mutation detection rate of 74% for combined germline and somatic mutation testing with a custom panel comprising 17 genes. This study, however, identified half of the mutations to be of germline and half of somatic origin [31]. This discrepancy could be explained by a high percentage of patients with seemingly sporadic tumors in our cohort. Average age of tumor onset for those 13 patients who received both germline and tumor analysis was 42 years and only one patient had proven malignant disease. None of these patients had a positive family history for PPGL-associated tumors.

While patients can benefit from combined tumor and germline analysis, this approach is generally hindered by both financial issues and organizational obstacles. We did observe such obstacles in our cohort for those patients of whom, due to anonymous inclusion in a multi-center trial, we could only analyze tumor tissue without having access to germline information. Interdisciplinary approaches, especially between pathologists and clinical geneticists, could aid in bringing together both germline and tumor sequencing results in order to provide attending physicians with comprehensive genetic data.

Commonly used commercial sequencing panels for cancers are usually designed to either be used in clinical genetics in search for predisposing germline variants (such as the TruSight Cancer panel used in our routine diagnostic approach) or for application in molecular pathology for identification of somatic (hot-spot) variants in tumor tissues with focus on FFPE samples. Our custom panel covers both, i.e., genes that are typically mutated in the germline in context of hereditary tumor diseases (such as *SDHx* or *FH*) and genes that are almost exclusively somatically mutated in PPGLs (such as *HRAS*, *ATRX*, *IDHx*, etc.). It is moreover suitable for testing of high-quality DNA from blood samples as well as low-quality DNA from FFPE tissue samples. Therefore, our PPGL panel can be applied to combined germline and tumor analysis, leading to a significant increase in detected mutations.

The mutational spectrum we observed was generally compliant with previously reported results [3,4,31]. We did find a total of 3/28 (10.7%) somatically *ATRX* mutated tumors, which is presumed to be associated with aggressive disease [22,39,40]. One of these patients had not developed any metastases at the time of investigation, one patient had recurrence of disease and for one patient, follow-up information was unavailable. The first patient additionally carried a pathogenic *SDHC* germline mutation, which is in line with the reported enrichment of *ATRX* mutations in *SDHx* mutated tumors [32]. The second tumor showed a somatic pathogenic *TP53* mutation in addition to the *ATRX* truncating mutation. *TP53* mutations are rare in PPGLs compared to most other tumor types. However, it has been proposed they could have a synergistic effect on other driver mutations, and it seems plausible that both the *ATRX* and *TP53* mutations affected tumor development in this patient. Different from *ATRX* mutations, however, *TP53* mutations in PPGL have not been associated with more aggressive disease. [3,22] The third patient in our cohort with an *ATRX* mutation was not identified to carry any other driver alterations and did only show a read frequency of 5.4% for the *ATRX* mutation. While sole *ATRX* mutations have been described to drive sporadic PPGL development, it could also be considered that in this patient a relevant second mutation might have escaped the analysis [32,39]. 

Copy number variants (CNVs), for instance, are typically not detected by NGS-based variant calling [26]. We therefore applied a CNV detection approach based on the sequencing data generated by our custom panel in order to complement single nucleotide variant (SNV) detection from both cryo-conserved tumor samples and paraffin embedded tissues. This approach detected both loss-of-heterozygosity and possibly disease-related mutations in several tumors. However, further improvements and validations would be needed in order to reliably apply this technique in a diagnostic setting. This might be of value, since array-CGH can be unreliable for highly degraded DNA from FFPE tumor samples. CNV-detection based on our NGS custom-panel data could therefore provide a feasible alternative for CNV detection in these cases.

Apart from possible CNV detection and combined tumor and germline analysis, our broad customized PPGL panel did show further advantages over limited diagnostic panels, such as possible detection of rare or unexpected genetic events. For example, we identified a somatic *IDH2* hot spot mutation (Arg172Gly) in a paraganglioma sample, as reported previously [21]. In addition, several variants of unknown significance were identified in PPGL candidate genes, allowing for further investigation of these potentially relevant genes. Considering steady research advances in underlying genetic mechanisms of PPGLs, the panel can easily be extended to further target regions. 

However, one limitation of common sequencing approaches, which also applies to our panel, is the identification of epigenetic events leading to PPGL development. In one patient in our cohort, who developed multiple tumors, no pathogenic mutation was identified by common sequencing approaches, since the underlying epigenetic event was hypermethylation of the *SDHC* promoter region, as was reported earlier [41]. This example emphasizes that even with broad diagnostic sequencing panels, unsolved cases with suspicion of hereditary disease should be directed to research approaches. Metabolic data can substantially aid diagnostics by guiding the search for genetic alterations that escape routine genetic testing [21,23,41].

Taking these results together, we propose an interdisciplinary approach for genetic analysis of PPGL that encompasses both tumor and germline sequencing and conclude that our custom designed PPGL panel provides a valid tool for the identification of SNVs from fresh frozen and paraffin-embedded tumor tissues. CNV detection can further be performed based on the data generated by our panel. Combined diagnostic and research approaches yield optimal results for comprehensive genetic characterization of PPGL patients.

## 4. Materials and Methods 

### 4.1. Patient Cohort and Genetic Testing Strategy

Sequencing of blood or tumor samples was conducted either within a diagnostic or a research setting after informed written consent was given by all patients at their respective centers of study inclusion. In a diagnostic setting, patients were initially seen by endocrinologists and referred to the Institute for Clinical Genetics for genetic counselling. Pedigrees were analyzed and germline analysis was initiated, covering PPGL core genes *SDHB, SDHC, SDHD, SDHAF2, MAX, FH, NF1, RET, TMEM127* and *VHL* either via Sanger-Sequencing and/or next generation sequencing using a commercially available gene panel (TruSight Cancer Panel, Illumina) [34,42]. This was complemented by custom array-CGH analysis for detection of copy number variations and multiplex ligation-dependent probe amplification (MLPA) (*SDHA*). If a causative mutation was identified, genetic counselling and predictive testing in families as well as appropriate clinical management and/or study inclusion were initiated. If tumor samples were available, cases with no identified causative germline mutations were analyzed using our dedicated PPGL multi gene panel comprising 84 genes. Additionally, blood or tumor tissue samples were provided in anonymized fashion by several centers from Madrid, Nijmegen, Munich and Bethesda under the clinical protocol of the Prospective Monoamine-producing Tumor Study (coordinator Graeme Eisenhofer). A number of samples reported here have already been published in a study with a different research focus [21]. Patients from the research cohort were either part of the Prospective Monoamine-producing Tumor Study (PMT study; https://pmt-study.pressor.org/) and/or Registry and Repository of biological samples of the European Network for the Study of Adrenal Tumours (ENS@T) with ethic approval given at each participating institution (ethic codes: EK 189062010 (Dresden), 2011-334 (Nijmegen), 173-11 and 379-10 (Munich), 15/024 (Madrid), 2011/0020149 Ref. n. 59/11 (Florence)), or enrolled under the IRB Protocol 00-CH-0093 (NIH/Bethesda/USA). Patients from the diagnostic cohort signed informed consent for genetic testing in accordance with the German Genetic Diagnostics Act (GenDG).

### 4.2. Next Generation Sequencing Analysis

DNA was isolated either from blood or from tumor tissue samples provided fresh or formalin fixed paraffin embedded. NGS analysis was conducted on a NextSeq or MiSeq sequencing instrument (Illumina Inc., San Diego, CA, USA) using the TruSight Cancer sequencing panel (Illumina Inc., San Diego, CA, USA) in the case of diagnostic analysis or the custom designed PPGL Panel (Illumina Inc., San Diego, CA, USA Appendix A) for research analysis. 

Resulting fastq sequence files were aligned to the human reference genome hg19 (GRCh37) using the Biomedical Genomics Workbench 5.0 (Qiagen, Hilden, Germany). In the case of diagnostic analysis, variants were called with a fixed ploidy algorithm with a required minimum frequency of 10 %, 3 reads supporting the variant and a required minimum coverage of 10 reads. For research analysis of tumor tissue or blood, variants were called using a low frequency variant detection algorithm (no assumption of known sample ploidy) with a required minimum frequency of 5% and otherwise similar settings. Variant calling was restricted to target regions as defined by the bed files of the TruSight Cancer panel (Illumina Inc., San Diego, CA, USA) and the PPGL panel. Variant classification was performed in accordance with the standards and guidelines of the American College of Medical Genetics and Genomics and the Association for Molecular Pathology (ACMG-AMP) [43].

### 4.3. CNV Calling

NGS-based calling of copy number variations (CNVs) was performed with the R (https://www.r-project.org/, [44]) package panelcn.MOPS [45], a CNV detection tool for targeted panel NGS data, set to default parameters. DNA obtained from blood samples of ten healthy individuals was sequenced with the PPGL panel and served as normal controls for CNV detection. The absence of CNVs in known PPGL susceptibility genes (*SDHA*, *SDHB, SDHC, SDHD, SDHAF2, MAX, FH, NF1, RET, TMEM127* and *VHL)* in controls was confirmed by array-CGH. Additionally, a sample with a heterozygous deletion of *NF1* as previously identified by array-CGH was included as a positive control for CNV detection. For each sample, log2 values, statistical parameters and copy numbers (CNx; x = number of copies) were exported into a csv file (Appendix A). Furthermore, for each sample, median log2ratios per gene were plotted for visualization (Appendix A).

### 4.4. Metabolite Analyses

Succinate and fumarate were extracted with methanol from fresh or FFPE tissue as previously reported [23] and analyzed by liquid chromatography tandem mass spectrometry (LC-MS/MS). Methodological details have previously been reported [21]. 

## 5. Conclusions

Comprehensive testing of tumor samples can improve diagnostics in PPGL patients. We propose parallel testing of blood and tumor tissue (fresh frozen or FFPE), accompanied by metabolite profiling, immunohistochemistry and additional analyses such as methylation detection to allow for better data interpretation. In the future, current NGS panel designs have to be updated to facilitate integration of newly discovered susceptibility genes or could even be replaced by exome sequencing. Patients can substantially benefit from integrating diagnostic and research approaches for molecular characterization of PPGLs. 

## Figures and Tables

**Figure 1 cancers-11-00809-f001:**
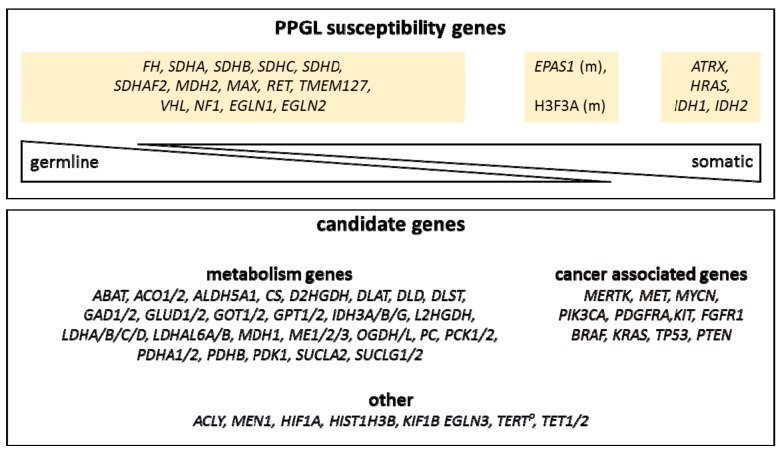
Schematic overview of the custom pheochromocytoma and paraganglioma (PPGL) panel. Known PPGL susceptibility genes and expected occurrences (germline, somatic) are depicted in the upper panel. Candidate gene categories are depicted in the lower panel. (m): mosaic, *TERT^P^: TERT promoter region.*

**Figure 2 cancers-11-00809-f002:**
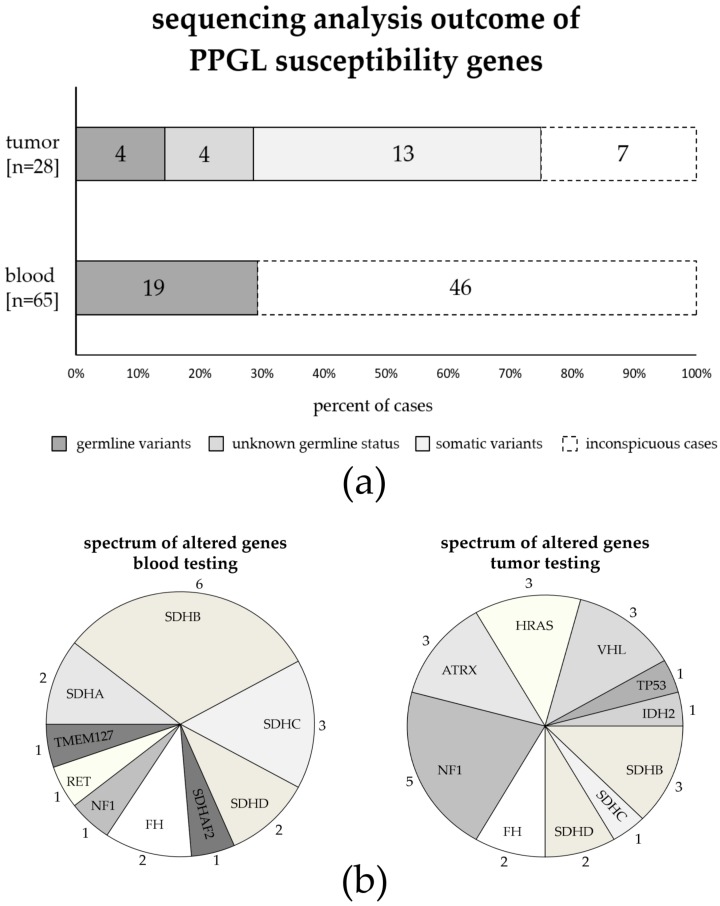
Summarized outcome of sequencing analysis of pheochromocytoma and paraganglioma (PPGL) susceptibility genes by diagnostic sequencing of blood samples in a clinical cohort of 65 patients and next generation sequencing analysis of 28 tumor tissues using our PPGL panel. Ten patients received both, diagnostic blood sequencing and tumor analysis, and are included in both cohorts. (**a**) We found (likely) pathogenic mutations in 21 of 28 tumors (75%), including three samples with more than one mutation (see Table 2); four of those patients had confirmed germline variants (dark grey); in four patients, germline status of the variant is unknown (grey); 13 cases had either confirmed somatic mutations or presumably somatic mutations based on the genes involved and/or the allele frequencies of the variants (light grey). Routine clinical blood testing in the clinical cohort of 65 patients identified 19 cases with germline mutations (29.2%). (**b**) Different spectrum of genes found to be mutated when performing blood testing (19 (likely) pathogenic variants identified in 19 of 65 blood samples) or tumor testing (24 (likely) pathogenic variants identified in 21 of 28 tumor samples). Numbers indicate how many variants were identified per gene.

**Figure 3 cancers-11-00809-f003:**
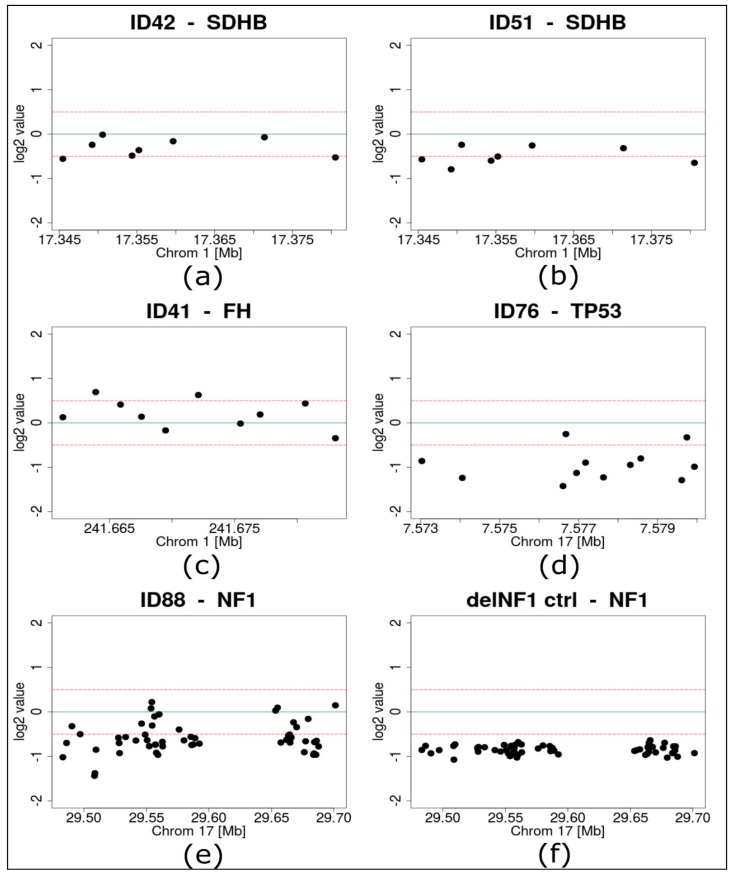
Results of next generation sequencing based copy number variant detection. (**a**) *SDHB* of case ID42, (**b**) *SDHB* of case ID51, (**c**) *FH* of case ID41, (**d**) *TP53* of case ID76, (**e**) *NF1* of case ID88, (**f**) *NF1* of a control sample with an experimentally proven *NF1* deletion; dots represent single targets of the respective genes, green lines highlight a log2 value of 0, dotted red lines mark log2 values of +0.5 and −0.5.

**Table 1 cancers-11-00809-t001:** Overview of (likely) pathogenic germline variants (ACMG classes 4/5) identified during routine testing.

ID	Diagnosis	Solitary/Multiple	AD/Gender	Family History	Gene	Nucleotide Change	Amino Acid Change	S:F Ratio in Tumor Tissue
ID45	PHEO	unknown	30/f	unknown	FH	c.434C > G	p.(Ser145*)	0.14
ID11	PHEO	solitary	59/m	unknown	FH	c.1431_1433dupAAA	p.(Lys477dup)	n.a.
ID32	PHEO	multiple	39/f	1 melanoma (AO 35)	NF1	c.6084+1G > A	p.?	n.a.
ID52	PHEO	multiple	47/f	inconspicuous	RET	c.1901G > T	p.(Cys634Phe)	5.5
ID56	HNP	unknown	23/m	unknown	SDHA	c.553_554insA	p.(Ala186fs)	n.a.
ID62	PGL	solitary	37/f	inconspicuous	SDHA	c.1338delA	p.(His447fs)	n.a.
ID35	PGL/HNP	multiple	52/f	1 melanoma (AO 48)	SDHAF2	c.232G > A	p.(Gly78Arg)	217.3
ID3	HNP	solitary	17/m	inconspicuous	SDHB CHEK2	c.649C > T c.1100delC	p.(Arg217Cys) p.(Thr367fs)	24.1
ID42	PGL	solitary	51/m	cancers (AO > 50)	SDHB	c.287-3 C > G	p.?	1472.7
ID4	PGL/HNP	multiple	23/f	cancers (AO > 50)	SDHB	deletion exon 3		n.a.
ID55	HNP	unknown	26/f	unknown	SDHB	c.806delT	p.(Met269fs)	n.a.
ID86	PGL	solitary	33/m	unknown	SDHB	c.649C > T	p.(Arg217Cys)	n.a.
ID19	HNP	solitary	36/f	cancers (AO > 50)	SDHB	c.725G > A	p.(Arg242His)	5908.3
ID63	PGL	solitary	48/m	cancers (AO > 50)	SDHC	c.397C > T	p.(Arg133*)	45.4
ID48	HNP	solitary	69/f	inconspicuous	SDHC	c.43C > T	p.(Arg15*)	n.a.
ID43	PHEO	solitary	50/m	cancers (AO > 50)	SDHC	c.379C > T	p.(His127Tyr)	795.5
ID34	HNP	solitary	34/m	2 PGLs	SDHD	c.53-2A > G	p.?	278.3
ID38	HNP	solitary	47/f	1 PGL	SDHD	c.49C > T	p.(Arg17*)	920.6
ID59	HNP	unknown	68/f	unknown	TMEM127	c.465_466insACTTG	p.(Ala156fs)	n.a.

AD: age at diagnosis, AO: age of onset, PHEO: pheochromocytoma, PGL: paraganglioma, HNP: head & neck paraganglioma, S:F ratio: succinate to fumarate ratio.

**Table 2 cancers-11-00809-t002:** (Likely) pathogenic variants (ACMG classes 4/ 5) identified in tumor samples.

	Diagnosis	AD/Gender	Gene	Nucleotide Change	Amino Acid Change	VAF (Tumor)	LOH	Somatic Status	Germline Testing *	S:F Ratio
ID80	PHEO	65/m	*ATRX*	c.1441G > T	p.(Glu481*)	5.4%	no	likely somatic	no	46.2
ID82	PHEO	31/f	*FH*	c.700A > G	p.(Thr234Ala)	82.0%	yes	germline	yes (targeted)	0.4
ID41	PHEO	37/m	*FH*	c.816_836del	p.(Ala273_Val279del)	92.3%	yes	germline	yes (targeted)	0.3
ID68	PHEO	66/f	*HRAS*	c.182A > G	p.(Gln61Arg)	56.8%	no	somatic	yes	66.3
ID1	PHEO	52/f	*HRAS*	c.182A > G	p.(Gln61Arg)	72.0%		somatic	yes	17.7
ID60	PHEO	27/f	*HRAS*	c.37G > C	p.Gly13Arg	26.0%	no	somatic	no	12.8
ID75	HNP	53/f	*IDH2*	c.514A > G	p.Arg172Gly	24.5%	no	somatic	yes (targeted)	5.3
ID73	PHEO	56/f	*NF1*	c.1540C > T	p.(Gln514*)	62.1%		unknown	no	16.6
ID79	PHEO	58/f	*NF1*	c.7798_7799insA	p.(Ser2601fs)	83.2%	yes	unknown	no	47.5
ID91	PHEO	50/m	*NF1*	c.205-1G > T	p.?	39.9%	no	unknown	no	unknown
ID92	PHEO	73/f	*NF1* *NF1*	c.1318C > T c.7549C > T	p.(Arg440*) p.(Arg2517*)	15.9% 33.2%	no no	likely somatic unknown	no	unknown unknown
ID51	PGL	56/m	*SDHB*	c.183T > G	p.(Tyr61*)	80.0%	yes	somatic	yes (targeted)	5178.2
ID42	PGL	51/m	*SDHB*	c.287-3C > G	p.?	85.4%	yes	germline	yes	1472.7
ID71	HNP	49/m	*SDHB*	c.724C > T	p.(Arg242Cys)	16.4%	no	likely somatic	no	24.7
ID43	PHEO	50/m	*SDHC* *ATRX*	c.379C > T c.2817del	p.(His127Tyr) p.(Ala940fs)	46.6% 65.8%	no	germline likely somatic	yes	795.5
ID69	HNP	27/f	*SDHD*	c.3G > T	p.(Met1Ile)	18.2%	no	somatic	yes	405.9
ID24	PGL	21/m	*SDHD*	c.337_340del	p.(Asp113fs)	41.5%		somatic	yes	1756.8
ID72	PHEO	66/f	*TP53* *ATRX*	c.817C > T c.4744_4745insA	p.(Arg273Cys) p.(Thr1582fs)	74.1% 61.2%		somatic unknown	yes	5.7
ID67	PGL	31/f	*VHL*	c.193T > G	p.(Ser65Ala)	8.5%	no	somatic	yes	24.1
ID66	PHEO	13/m	*VHL*	c.193T > G	p.(Ser65Ala)	17.4%	no	somatic	yes	21.4
ID78	PHEO	47/f	*VHL*	c.500G > A	p.(Arg167Gln)	49.6%	no	unknown	no	17.4

* Information about germline status was either available from routine germline testing in the patients included from our clinical cohort (indicated with “Germline testing yes”) or due to targeted Sanger sequencing of blood samples (“yes (targeted)”). ACMG: American College of Medical Genetics, AD: age at diagnosis, LOH: loss of heterozygosity PHEO: pheochromocytoma, PGL: paraganglioma, HNP: head and neck paraganglioma, S:F ratio: succinate to fumarate ratio, VAF: variant allele frequency.

**Table 3 cancers-11-00809-t003:** Overview of variants of unknown significance in candidate genes.

ID	Gene	Nucleotide Change	Amino Acid Change	VAF (Tumor)	gnomAD het/hom	SIFT	PolyPhen	COSMIC	dbSNP	Pathogenic Variants
ID61	*ATRX*	c.157A > C	p.(Asn53His)	25.2%	0/0	tolerated	probably damaging	-	-	no
ID78	*FGFR1*	c.2104C > A	p.(Pro702Thr)	49.6%	0/0	damaging	probably damaging	1 × lung	-	yes (*VHL*)
ID69	*FH*	c.593C > T	p.(Ala198Val)	7.3%	0/0	damaging	probably damaging	-	-	yes (*SDHD*)
ID68	*GPT*	c.628G > A	p.(Glu210Lys)	46.0%	0/0	damaging	probably damaging	1 × large intestine	rs1366336459	yes (*HRAS*)
ID79	*HIST1H3B*	c.131C > A	p.(Pro44Gln)	52.3%	0/0	damaging	n.a.	-	-	yes (*NF1*)
ID41	*OGDHL*	c.1340A > G	p.(Tyr447Cys)	51.1%	32/0	damaging	probably damaging	-	rs148307090	yes (*FH*)
ID82	*PCK2*	c.463C > T	p.(Arg155Cys)	50.9%	30/0	damaging	probably damaging	-	rs141787425	yes (*FH*)
ID71	*PDHB*	c.520G > A	p.(Val174Met)	47.7%	1/0	damaging	probably damaging	-	rs760966357	yes (*SDHB*)
ID88	*TET1*	c.382G > C	p.(Val128Leu)	26.1%	2/0	tolerated	benign	-	rs142008363	no

VAF: variant allele frequency, het: heterozygous, hom: homozygous, SIFT: Sorting Intolerant From Tolerant (*in silico* variant effect prediction), COSMIC: the Catalogue Of Somatic Mutations In Cancer.

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
