# Peer review of "Optimizing Genetic Workup in Pheochromocytoma and Paraganglioma by Integrating Diagnostic and Research Approaches"

_cancers, 2019, doi:10.3390/cancers11060809_

Round 1
Reviewer 1 Report
Gieldon et al. describe a cohort analysis of 93 patients with two types of rare neuroendocrine tumors: pheochromocytomas and paragangliomas (PPGL), where two types of next-generation sequencing analysis panels (Illumina TruSight and their own Illumina PPGL panel). They show standard clinical testing (using the Illumina TruSight panel) found that 19 of 65 cases (29.2%) had germline mutations in known PPGL-susceptibility genes, supporting the observation of previous studies that approximately 1/3rd of patients with the disease have a germline-susceptibility variant.
The novelty of their work lies in the application of a targeted panel to 28 patients with the disease for which they have banked tumor sample. They observe a large number of patients (75%) have mutations in the tumor samples in known PPGL associated genes.
Major Comments
A. My one major concern with the manuscript is the lack of germline testing in the 28 samples. I understand that this is likely now impossible but without this data, some of the strong language around interpretation of the results needs to be attenuated. For example, assertation that the analysis of these 28 patients “revealed disease-causing variants” (line 59) in known PPGL-susceptibility genes in 75% of cases. I do not think the authors can make this claim as they only performed germline analysis in a small number of the patients (not clear how many see comment below) and found germline variants in only 2 samples (Table 2). For the remaining 19 somatic or possibly-somatic variants, how can the authors assert that they are disease causing? While I agree that it is likely they are playing a role in the patient’s disease, without knowing if they were both in the germline and pathogenic, I do not feel it is possible to say they are causing the disease. The possibility that other germline or driver mutations in the tumor cannot be excluded. The patients also may have had mosaic mutations in these genes, which lead to tumor formation. Or as you say, they could have been
B. The CNV analysis section is particularly weak. You had array data on a large number of these patients yet no comparison between array data and the results of the PPGL panel are made. I am not suggesting you do this, as perhaps arrays were not run on the 28 tumor samples, but that would have supported the validity of your proposal to use the panel for CNV analysis. There is no description of the analysis methodology used to call CNVs from you panel – this must be rectified. For example, how was the log2 ratio generated – is it tumor and matched normal? Are you using depth of coverage or VAF? Finally, figure 3 includes a known positive control but no negative control. In focusing just on the genic region of interest, it is not possible to get a sense of whether all other genes in the panel were copy neutral. Please create a plot where all genes are shown for each sample, indicating the gene you are claiming is deleted. You can then add the zoomed in regions you are currently showing in Figure 3.
C. I would like to have seen more of a contrasting between your new panel and the trusight panel. For example, of the 84 genes in your panel, 16 are present in the TruSight panel (FH, HRAS, KIT, MAX, MEN1, MET, NF1, PTEN, RET, SDHAF2, SDHB, SDHC, SDHD, TMEM127, TP53 and VHL). Please could you discuss the fact that of the 68 new genes you tested, only 4 of the 21 patients had pathogenic changes in two genes not in the Illumina panel (ATRX and FH). Given the other 17 patients with pathogenic variants would have been identified by the TriSight panel does that imply the new PPGL panel is of limited additional benefit? Please include some of these observations in your results and compare and contrast the advantages of your panel over the current clinical care panel in more detail in the discussion.
Minor Comments
1. Not being familiar with the term “PPGL” I am not sure it is wise to not indicate in the title of the manuscript what PPGL stands for or that you are examining neuroendocrine tumors. Please revise the title.
2. The results section is a little confusing in regard to which samples had a germline sample available and which did not. The cohort is split into a group of 10 patients that had clinical testing and 18 patients without clinical testing. Please indicate this in Table 2 (would help readers link the results in the table to Figure 2). Please also indicate which patients were tumor only vs. germline and tumor.
3. Please define how you consider a variant pathogenic in table 2. Are these variants known in clinvar (if yes, include their clinvar accession in the table) and have they been assessed by any standard (e.g. ACMG or AMP guidelines).
4. This may be related to the lack of clarity on germline vs tumor testing in the patients, but I do not think your results support the claim in the discussion on line 327, that one third of mutations were found in the germline. This contradicts Table 2 where only 2 samples are indicated as being germline. Please either correct this statement or make the data driving this observation clearer.
Author Response
Major Comments
Point A: My one major concern with the manuscript is the lack of germline testing in the 28 samples. I understand that this is likely now impossible but without this data, some of the strong language around interpretation of the results needs to be attenuated. For example, assertation that the analysis of these 28 patients “revealed disease-causing variants” (line 59) in known PPGL-susceptibility genes in 75% of cases. I do not think the authors can make this claim as they only performed germline analysis in a small number of the patients (not clear how many see comment below) and found germline variants in only 2 samples (Table 2). For the remaining 19 somatic or possibly-somatic variants, how can the authors assert that they are disease causing? While I agree that it is likely they are playing a role in the patient’s disease, without knowing if they were both in the germline and pathogenic, I do not feel it is possible to say they are causing the disease. The possibility that other germline or driver mutations in the tumor cannot be excluded. The patients also may have had mosaic mutations in these genes, which lead to tumor formation. Or as you say, they could have been.
Response A: We thank reviewer 1 for thorough reading and discussion of our manuscript. We completely agree that it would be important to establish the germline status for all variants identified in the 28 patients who received tumor analysis. Unfortunately, we did not have access to suitable tissues for germline testing in all of the cases. However, in 14 cases germline testing was conducted. Ten of the 28 patients were from our clinical cohort, who received routine germline analysis before tumor testing. In additional four patients of the research cohort, targeted germline testing for the mutations found in the tumor was performed. Of the remaining 14 cases, six had no pathogenic variants in the tumor tissue. In overall 8 cases of our tumor cohort, germline status of identified mutations could unfortunately not be confirmed due to lack of normal tissue samples. Nevertheless, several of those mutations were identified in genes reported to be typically mutated somatically in PPGL and/or had a variant frequency indicative for a somatic mutation. To give a clear overview of the germline testing status of all 28 tumor samples, we added a column to table 2. We also altered figure 2 and the caption of Figure 2, in order to clarify this.
We also, thanks to the reviewers comment, correct a mistake in the number of confirmed germline variants in table 2, which were altogether 4 confirmed germline variants.
We additionally rephrased line 59 from “disease-causing” to “pathogenic or likely pathogenic” in order to attenuate as was suggested by reviewer 1. We used the term disease-causing variant in the sense of identification of a pathogenic or likely pathogenic mutation in a known disease-causing gene. To avoid misunderstandings, we now use the terms likely pathogenic or pathogenic to describe variants in accordance with ACMG guidelines in order to rate the variants by ACMG-classes without commenting on their impact on disease development. We also changed the term “disease-causing” in lines 141 and 310 with regards to pathogenic germline mutations, in order to draw a clearer line. We further rephrased line 331 and 407 to further attenuate.
We consider the lack of integration of genetic information from germline and tumor testing one of the major weaknesses in PPGL-research and diagnostics in general. For precisely this reason, we argue in our discussion (from line 346) that interdisciplinary approaches, integrating germline and somatic testing, need to be set in place for PPGL patients. We do see the benefit of this in those patients of whom we did have both germline and tumor tissue. To emphasize
this, we added a line describing one of the organizational hazards we encountered in bringing together germline and tumor testing (from line 374).
Point B: The CNV analysis section is particularly weak. You had array data on a large number of these patients yet no comparison between array data and the results of the PPGL panel are made. I am not suggesting you do this, as perhaps arrays were not run on the 28 tumor samples, but that would have supported the validity of your proposal to use the panel for CNV analysis. There is no description of the analysis methodology used to call CNVs from you panel – this must be rectified. For example, how was the log2 ratio generated – is it tumor and matched normal? Are you using depth of coverage or VAF? Finally, figure 3 includes a known positive control but no negative control. In focusing just on the genic region of interest, it is not possible to get a sense of whether all other genes in the panel were copy neutral. Please create a plot where all genes are shown for each sample, indicating the gene you are claiming is deleted. You can then add the zoomed in regions you are currently showing in Figure 3.
Response B: We apologize sincerely that we mistakenly not included the description of CNV detection in the manuscript. We added the section describing the analysis methodology to the material and method section. In supplementary table 3, log2ratios for all genes and samples are given with the information, if a copy number change was called by the software or not. Supplementary table 3 was updated accordingly and now also contains CNV data from the ten controls. Additionally, we provided a supplementary file containing plots of CNV data from all 84 analyzed genes for each sample, the positive control with the heterozygous NF1 deletion and our ten normal controls.
Indeed, it would be interesting to compare arrayCGH data with our NGS panel sequencing data, however we indeed did not perform arrayCGH on tumor samples. One of the reasons is that for FFPE samples, arrayCGH is usually not applicable. This is one of the motivations to test if our NGS panel can be used for CNV-calling.
Point C: I would like to have seen more of a contrasting between your new panel and the trusight panel. For example, of the 84 genes in your panel, 16 are present in the TruSight panel (FH, HRAS, KIT, MAX, MEN1, MET, NF1, PTEN, RET, SDHAF2, SDHB, SDHC, SDHD, TMEM127, TP53 and VHL). Please could you discuss the fact that of the 68 new genes you tested, only 4 of the 21 patients had pathogenic changes in two genes not in the Illumina panel (ATRX and FH). Given the other 17 patients with pathogenic variants would have been identified by the TriSight panel does that imply the new PPGL panel is of limited additional benefit? Please include some of these observations in your results and compare and contrast the advantages of your panel over the current clinical care panel in more detail in the discussion.
Response C: We agree with Reviewer 3 that many of the genes found to be mutated in tumor samples are also present on the TruSightCancer panel. However, benefits of the custom panel are not only based on the additional genes included in the panel, but also on the applicability to different tissues, including FFPE tumor samples. Indeed many of the mutations identified in tumor tissues of the 28 patients would still not have been identified by the TruSight panel, since it is not applicable to FFPE samples (see below).
Regarding the tested genes, the panel allows for identification of unexpected mutations such as the IDH2 mutation in ID71, which, to our knowledge was the first IDH2-mutation described in PPGLs so far and which we reported earlier (reference 21). We also identified variants in several candidate genes, as discussed starting in line 410. Since we only applied the panel to 28 samples so far, we do believe inclusion of candidate genes could lead to further interesting findings within future research projects.
Importantly, we consider its applicability to both tumor and germline testing one of the major advantages of our custom panel. Commonly used pre-designed sequencing panels for cancers are usually designed to either be used in human genetics in search for predisposing germline variants and applicable for blood (such as the TruSight Cancer panel used in our routine diagnostic approach) or for application in somatic analysis of tumors in molecular pathology. For example, although HRAS and TP53 are included in the TruSightCancer panel, this panel cannot be used on FFPE tumor samples and since mutations in these genes in PPGLs are usually somatic, these cases would not have been solved by the panel. Indeed, none of the 65 patients in our cohort tested by the TrueSightCancer panel in the germline did show mutations in HRAS, ATRX, or TP53. The better highlight the advantages of our panel in the context of tumor testing, we added the spectrum of genes found to be mutated when testing tumor tissue compared to routine germline testing in figure 2b. In addition, we highlighted this advantage of our panel by adding a section to the discussion starting in line 380-388.
We also discussed the usefulness of our panel for CNV detection, starting in line 411, which is a further advantage of our panel.
Minor comments
Point 1: Not being familiar with the term “PPGL” I am not sure it is wise to not indicate in the title of the manuscript what PPGL stands for or that you are examining neuroendocrine tumors. Please revise the title.
Response 1: The title was changed accordingly.
Point 2: The results section is a little confusing in regard to which samples had a germline sample available and which did not. The cohort is split into a group of 10 patients that had clinical testing and 18 patients without clinical testing. Please indicate this in Table 2 (would help readers link the results in the table to Figure 2). Please also indicate which patients were tumor only vs. germline and tumor.
Response 2: A column containing this information was added in table 2 and figure 2 was revised (please also see Response A).
Point 3: Please define how you consider a variant pathogenic in table 2. Are these variants known in clinvar (if yes, include their clinvar accession in the table) and have they been assessed by any standard (e.g. ACMG or AMP guidelines).
Response 3: All variants were assessed in accordance with ACMG-AMP guidelines. The information was added in the methods section in line 472. In addition, the headings of table 1 and 2 was adapted to include this information.
Point 4: This may be related to the lack of clarity on germline vs tumor testing in the patients, but I do not think your results support the claim in the discussion on line 327, that one third of mutations were found in the germline. This contradicts Table 2 where only 2 samples are indicated as being germline. Please either correct this statement or make the data driving this observation clearer.
Response 4: We corrected a mistake in table 2, were 2 mutations were wrongly not be indicated as germline. We furthermore changed table 2 to give clarity about germline testing and also added information throughout the text were appropriate. In total, combined tumor and germline
sequencing was done in 14 cases, including 10 cases that underwent routine diagnostics (in 9 of them we identified a somatic or germline mutation) and 4 cases in which targeted germline testing was performed after tumor sequencing results were available. Altogether, we had 13 patients with mutations and combined germline and tumor testing. 4 of these 13 had a proven germline mutation, the rest had proven somatic alterations. This is how approximately 1/3rd was calculated. We clarified this also in the discussion in line 362-364 (former line 327).
Reviewer 2 Report
This paper describes the potential benefit of parallel genetic testing of peripheral blood and tumor tissue for the management of patients with pheochromocytomas and paragangliomas (PPGL). The results are interesting and manuscript is very well written. I have only a few comments.
1. In the analysis of 28 PPGL tumors with a PPGL custom panel, disease-causing variants were found in 21 tumors (75%). Was there any difference in the clinical characteristics between ones with and without those variants, for example, age of onset, presence or absence of metastasis, and metanephrine levels?
2. Of 65 patients with PPGLs, five had multiple PPGL tumors. Did the authors perform sequencing analysis on multiple tumors from each case? It would be interesting to see if different tumors from a patient had different somatic mutations in PPGL susceptibility genes.
Author Response
Minor comments
Point 1: In the analysis of 28 PPGL tumors with a PPGL custom panel, disease-causing variants were found in 21 tumors (75%). Was there any difference in the clinical characteristics between ones with and without those variants, for example, age of onset, presence or absence of metastasis, and metanephrine levels?
Response 1: We thank reviewer 2 for revising and discussion of our manuscript. We did not see a mentionable difference between those patients in whose tumor samples mutations were identified and those in whose samples we did not find a mutation. Patients without mutations include both pheochromocytoma and paraganglioma patients, both male and female patients, and age of onset ranges broadly from 26 to 82 years. We therefore decided not to go into further detail in the manuscript. We did, however, observe that in all three families with suspicious family history for hereditary PPGL, pathogenic germline mutations were identified and therefore stated this in lines 127-129. We included all available clinical data in supplementary table 1.
Point 2: Of 65 patients with PPGLs, five had multiple PPGL tumors. Did the authors perform sequencing analysis on multiple tumors from each case? It would be interesting to see if different tumors from a patient had different somatic mutations in PPGL susceptibility genes.
Response 2: We absolutely agree with reviewer 2 that this would be very interesting to do. Unfortunately, different tumors were available for sequencing only in the one single case in which an SDHC promotor methylation was identified, as is referenced in line 422-424.
Round 2
Reviewer 1 Report
Thank you for taking the time to address all of my comments and concerns - the manuscript is greatly improved and will be of significant interest to those working in the field of PPGL research and beyond.